# Investigation of the Residual Mechanical and Porosity Properties of Cement Mortar under Axial Stress during Heating

**DOI:** 10.3390/ma14081944

**Published:** 2021-04-13

**Authors:** Zhifei Gao, Linbing Wang, Hailu Yang

**Affiliations:** 1National Center for Materials Service Safety, University of Science and Technology Beijing, Beijing 100083, China; b20160403@xs.ustb.edu.cn (Z.G.); Yanghailu@ustb.edu.cn (H.Y.); 2Joint USTB Virginia Tech Lab on Multifunctional Materials, USTB, Virginia Tech, Department Civil & Environmental Engineering, Blacksburg, VA 24061, USA; 3Research and Development Center of Transport Industry of New Materials, Technologies Application for Highway Construction and Maintenance, Beijing 100088, China

**Keywords:** cement mortar, X-ray CT, porosity, residual compressive strength, preload

## Abstract

The preload load on concrete during heating is considered to cause a ‘densification’ of cement mortar which led to the increased compressive strength. In order to assess the influence of coupled load and heating effects on porosity characteristics of concrete, the porosity of mortar after mechanical and thermal loading was measured by X-ray computed tomography (X-ray CT). The preload at pre-stress ratios of 0, 0.2, 0.4, and 0.6 (ratio of stress applied to the specimen to its compressive strength at room temperature) were applied on mortar specimens during heating. The residual compressive strengths of the heated and stressed mortar specimens were tested after cooling to room temperature. Combined analyses of the residual compressive strength test results and porosity test results, it shows that the porosity of the specimens under the coupled stressing and heating conditions were slightly lower than that under the unstressed conditions; however, the conclusion that the increase of compressive strength of stressed mortar was caused by the ‘densification’ of cement paste was insufficient. The preload reduced the cracks in the mortar, especially the crack induced due to the thermal mismatch in aggregates and hardened cement paste (HCP), and this may account for the increased compressive strength of stressed mortar.

## 1. Introduction

Exposure to fire is considered to be the most destructive process for concrete [1]. Physical and chemical transformations take place in concrete under fire, which results in a remarkable decrease of compressive strength [2,3,4]. Experimental tests on concrete cored from a fired building indicated that the residual compressive strength of concrete was only 30% of its original strength, which significantly affected the safety of the structures [5].

Many researchers have conducted experiments on the compressive strength of concrete [4,6,7,8,9,10,11,12,13,14]. The investigations indicate that temperature has a great influence on the compressive strength of concrete, and the compressive strength decreases with the increase of temperature. However, in actual concrete structures, the concrete members are stressed by the designed load and constrained by other members. Therefore, concrete in structures exposed to high temperatures is practically always heated under stress [15,16], and the thermal behaviors of stressed concrete are different from those of unstressed concrete [16,17]. The test on stressed concrete during heating has been employed by many researchers.

In the stressed test, the concrete specimen is subjected to preload before and throughout the heating process [18], and the most obvious difference between the stressed and unstressed concrete is that the compressive strength is higher when the specimen is stressed during heating than that of unstressed concrete during heating, which was first reported by Malhotra [19,20]. In order to investigate the effect of high temperature on stressed concrete, Abram [21], Khoury [22], and Schneider [23] conducted experimental tests on the stressed and unstressed concrete and came to the same conclusion as Malhotra [19,20]. Shi et al. [24] carried out tests on concrete with different pre-stress ratios at a maximum pre-stress ratio of 0.6. The test results showed that the preload increased the compressive strength and elastic modulus. The same conclusion was reached by Tao et al. [25], when a preload at a pre-stress ratio of 0.2 was applied on self-compacting concrete cylinder specimens. Kim et al. did a series of experiments on the stressed concrete [7,16,26], and carried out experiments on the compressive strengths of light-weight and normal concrete at high temperature at different pre-stress ratios [16]. The results showed that the light weight concrete had a higher compressive strength, and Kim suggested that the stress applied to the concrete during heating should be controlled at less than 40% of the concrete strength at room temperature. Castillo [27], Phan and Carino [28], and Fu et al. [29] performed tests on the compressive strength of high strength concrete with a preload of 40% of the compressive strength at room temperature during heating. The test results confirmed that the preload led to smaller relative strength loss. Based on a review of available experimental data, the constitutive relationship of stressed normal and high strength concrete at high temperatures was proposed by Aslani [18].

In the above-mentioned pieces of literature, preload was found beneficial to concrete compressive strength retention at and after high temperature. However, there was little difference in compressive strength and stiffness for the stressed and unstressed concrete tested under uniaxial compression [15].

As the main component of concrete, the performance of cement mortar and cement paste stressed during heating was investigated. Khoury et al. [22] defined the load induced thermal strain (LITS) and measured the LITS of cement paste and concrete at different pre-stress ratios. The results indicated that the difference between the LITS of concrete and cement paste was non-linearly related to temperature. Fu et al. [30] compared the compressive strength of cement paste and cement mortar stressed during heating at high temperature. The compressive strength reduction of mortar specimens was much serious than that of cement paste. In the experimental test on concrete, which has a max aggregate size of 3.15 mm, the stressed specimens showed a significant increase in the uniaxial strength and initial elastic stiffness [31], the stressed cement mortar produced a similar relative strength loss of concrete. However, Sarshar and Khoury [32] pointed out that the residual compressive strength of cement paste specimens heated under preload was lower than that of specimens without stress during heating.

The reasons that cause the improvement in compressive strength of concrete stressed during heating are the following: (1) preloading in compression reduces concrete damage (micro-cracking) during the heating process [33]; (2) the cracks in preloaded specimens are not free to develop, and the failure process is, therefore, slower [27]; (3) The compressive stress in the matrix should be unloaded first, then the tensile stress can be established, and micro-cracks are formed when the aggregate expands [34]; (4) the ‘densification’ of cement paste may lead to a decrease in porosity relative to the unloaded specimen, while the pre-compression provided by load can reduce the tensile stress in the specimens, especially during cooling [29,35].

The reasons for the increase in compressive strength of stressed concrete during heating are still perplexing. The hydration products of the cement paste decompose at high temperature which result in the reduction of concrete compressive strength and the coarsening of cement mortar porosity. However, the effect of preload loaded during heating on the porosity structure of concrete was rarely studied. At the same time, conclusions on the effect of preload loaded during heating on mortar compressive strength are also inconsistent. In this study, the experimental tests on cement mortar performance after being heated to different temperatures were conducted. The cement mortar was heated under different pre-stress ratios. The residual compressive strength of cement mortar was tested, and the porosity characteristics of cement mortar under different conditions were compared using an X-ray CT technique.

## 2. Experimental Program

### 2.1. Raw Material and Specimen Preparation

In order to investigate the performances of cement mortar after high temperature, the ordinary cement mortar was employed. The cement was produced by Tangshan Huayu cement plant. The cement mortar was prepared according to the “Method of Testing Cements—Determination of Strength (GB/T17671-1999)” [36]. In this study, P42.5 ordinary Portland cement, river sand, and tap water were adopted. The mixing ratio of water: cement: sand was 1:2:5. The fineness modulus of sand was 2.03. The size of the mortar specimens was 40 × 40 × 40 mm^3^. The mortar was cast into the mortar molds and kept at room temperature for curing. After 24 h, the mortar specimens were demolded and moved to a 20 ± 2 °C and 95% ± 5% relative humidity controlled room for a curing age of 28 days.

### 2.2. Temperature Regimen

The temperatures for the test were 20, 200, 400, and 600 °C. An electrical furnace with a maximum temperature of 700 °C was selected. The furnace was installed on a universal testing machine which was shown in Figure 1 so that the cement mortar specimen could be heated and loaded at the same time. The loading instrument adopted the CSS-1110 electronic universal testing machine produced by Changchun Institute of Testing Machines, Ministry of Mechanical and Electrical, Changchun, China.

In order to ensure that the center temperature of cement mortar specimens reached the target temperature and the temperature of specimens was uniform, the central temperature of mortar specimens was measured when the specimens were heating. For the selected high temperatures, three type-K chromel-alumel thermocouples were installed in the center of three cement mortar specimens for the center temperature measurement. As the specimens were heating, the temperature of the furnace center and the specimen center were measured at the same time. The test results showed that the central temperature of the specimens reached the target temperature within 30 min for the selected temperatures.

In order to avoid spalling, when the target temperature exceeded 400 °C, the heating process was divided into different phases: set the 400 °C as the target temperature and heated for 20 min at first, then increased by 100 °C for every 10 min. Keep the temperature for 30 min when the specimen temperature reached a uniform status. Then, the furnace is turned off and the door is opened to cool down the specimen temperature to room temperature naturally. The whole time of the specimens exposure to high temperature was near or more than one hour, especially at 400 °C and the temperature after that.

### 2.3. Test Procedure

In order to investigate the properties of cement mortar stressed during heating, the pre-stress ratio of 0, 0.2, 0.4, and 0.6 were selected and the test conditions were shown in Table 1. In the beginning, the compressive strength of cement mortar at room temperature was tested and the average bearing capacity of the tested mortar was 78.9 kN. Therefore, the forces of 16, 32, and 48 kN were corresponding to the preload of pre-stress ratios 0.2, 0.4, and 0.6 respectively. The preloads were loaded before heating at the loading rate of 0.5 mm/min, and kept the loading position of the test machine unchanged during the heating and cooling periods.

X-ray scanning and residual strength of the specimens were continued when the specimens cooled to room temperature in 24 h. The loading rate for the residual compressive strength of cement mortar was 0.5 mm/min too, and three specimens were tested for each condition and the average strength of the specimens was used.

For a conditioned specimen, each one was designated with a letter and numbers. The letter C and G stood for the room temperature and the high temperature. The number after the letters was the temperature that the specimen tested at. The third number was the pre-stress ratio that the specimen was applied on.

### 2.4. Specimen Preparation for CT and SEM Test

X-ray CT is a non-destructive technique for detecting the internal structure of matter [35,37], and widely used to study the porosity of cement mortar. In this study, in order to investigate the effect of preload loaded during heating on the pore character of cement mortar, the investigation of cement mortar porosity was carried out. As the test specimens have cooled down, X-ray CT scanning of cement mortar specimens using a high-resolution X-CT system of nanoVoxel-2000 produced by Sanying Precision instruments Co., Ltd. Tianjin, China. These tests were carried out in the Beijing University of Technology. The ray source voltage was 150 V, and the current was 140 A. The width and height of the pixel were both 0.064 mm. Select cross sections in X, Y, and Z directions to investigate the effect of heat treatment on the porosity of cement mortar. The Z direction is the loading direction of the specimen in Figure 1.

In order to eliminate the edge hardening during the analysis, 600 slices in the middle of the samples were selected in the analysis. The Avizo software was used for the 3D reconstruction by the 2D mortar slices.

A scanning electron microscope (ULTRA55 SEM, Carl Zeiss, Oberkochen, Germany) was used to observe the microstructure of the specimens. Small pieces which were no more than 5 mm in thickness were selected from the compressive failure specimens for the scanning electron microscope test. The specimens were further treated through drying, vacuum pumping, and gold coating.

## 3. Test Results

In the experiments, preload was applied to the mortar specimens before they were heated, and the loading position of the testing machine was kept constant during heating and cooling. One group of the specimens after heating was shown in Figure 2. In the figure, for example, the number 2 in the specimen code G600-2 represented the second group specimen. All the heated specimens can still maintain integrity. Thermal cracking only can be observed on the specimen G400-0.6-2 by the naked eye, so this group was selected as a representative. A color change of the mortar specimens from normal to pink was observed.

### 3.1. Areal Porosity of Cement after Thermal Treatment

X-ray CT technology was widely used in the investigation of cement porosity [37,38,39,40]. The deterioration of cement materials at high temperature is usually related to the drastic change of its microstructure. The pore structure of cement mortar that post heat was investigated by X-ray CT [41]. The investigations showed that X-ray CT technology is an important method to evaluate the porosity of mortar which was used in this study. The cross sections of mortar in X, Y, and Z directions were scanned and three slices in the front, middle, and back of the specimens in each direction were selected. When the CT scan was performed, the specimens were still kept the same direction as when heated, thus in the CT images, the porosity in Z direction is the porosity of horizontal section.

The attenuation coefficient of X-ray is closely related to the phase density, which results in the difference in gray value in CT slices [37,40]. The phases were separated from each other by the grayscale division [37].

In this study, the cement mortar was idealized as a two-phase microstructure, i.e., pore and solid phases. The dark area represented the pore, while the bright area represented the sold phases which was consisted of the hydration of cement paste, un-hydrated cement paste and sands. The CT image of the middle cross-section in X-direction shown in Figure 3 as an example.

In Figure 3, it can be seen that cement mortar has few initial pores at room temperature. With the increase of temperature, the hydration productions decomposed and cement mortar coarsened. Due to the influence of the preload, there was a significant crack in Figure 3 which was corresponding to the condition of G400-0.6.

In order to quantitatively describe the influence of thermal effect on the porosity of cement mortar, the areal parameter *D*_S_ is introduced in Equation (1) [42], which is the percentage of face voxels to total face voxels.
(1)$$D_{s}=\frac{S_{\text{pore}}}{S_{\text{total}}}\times 100\%$$
where *D*_s_ is the areal porosity of cement mortar, *S*_pore_ is the areal voxel numbers of pore and *S*_total_ is the entire cross-section areal voxel numbers. The 2D CT slices which were selected of the front, middle, and back cross-sections in the three sections were used to calculate the areal voxel number. The average *D*_s_ of the three selected slices in each direction was shown in Figure 4.

It can be seen that the areal porosity of the cement mortar increased with the increasing temperature. The *D*_s_ of G400-0.6 was the highest which was mainly caused by the cracks. The *D*_s_ at 600 °C was lower than that at 400 °C which may be caused by the nature porosity of the individual. The variation of the areal porosity in the different direction was not significant. Therefore, it can be inferred that the temperature of the whole model was uniform.

### 3.2. Volumetric Porosity of Cement Mortar Post Thermal Treatment

The Avizo software was used for the 3D reconstruction by the 2D mortar slices. Figure 5 showed the 3D reconstruction images of the cement mortar after being heated to different temperatures. In the 3D images, the dark red region represented the pores, while the red frame was the analysis area. Before thermal treated, there are pre-existing pores in cement mortar. With the increasing temperature, the hydration productions decomposed and more pores were observed.

The volume parameter *D*_v_ is introduced in Equation (2) [41]. The ratio of volume porosity prime number to total volume prime number is defined to evaluate the influence of thermal effect on the volume porosity of cement mortar.
(2)$$D_{v}=\frac{V_{\text{pore}}}{V_{\text{total}}}\times 100\%$$
where *D*_v_ is the volumetric porosity of the cement mortar, *V*_pore_ is the volumetric voxel numbers of pores and *V*_total_ is the entire specimen volumetric voxel numbers. The *D*_v_ of cement mortar at different temperature was shown in Figure 6.

Under the unstressed conditions, due to the dehydration of the ettringite and the loss of physically bound water, there was a slight increase of porosity after the mortar was heated at 200 °C [7,30]. However, when heated at 400 °C and the temperature after that temperature, the porosity of mortar increases significantly. That was because the decomposition of CSH accelerated dramatically after 400 °C, and the dehydration of CH mainly occurred above 400 °C [43].

Under the stressed condition, the porosity of stressed mortar was increased with the increasing temperature too. Comparing the porosity of mortar under stressed conditions and unstressed conditions at the same temperature, the difference of porosity was not obvious. The porosity of the specimens under the stressed condition was slightly lower than that of the specimens under the unstressed condition. However, the porosity of G400-0.6 was the highest in all the conditions, which was due to the crack induced in the heating.

### 3.3. Strength of the Cement Mortar after High Temperature

After the CT scan test, the test on the residual compressive of cement mortar after thermal treatment was carried out. The specimens after the compressive strength were shown in Figure 7.

The color change of mortar specimens from normal to pink and the coarsening of the porosity structure could be more clearly observed. With the increasing temperature, the compressive strength decreased and the specimens damaged seriously after the compressive strength test. The higher the test temperature, the more serious the damage.

The compressive stress–strain curves of mortar after high temperature were shown in Figure 8. It can be seen that in Figure 8a, the compressive stress–strain of mortar at room temperature was near linear before the peak strength. After the peak strength, the stress of mortar specimen decreased sharply. After heated, due to the decomposition of the hydration products, there was a significant decreasing on the peak strength and peak strain (the strain at the peak strength). The slope of the stress–strain curves (that corresponding with the tangent modulus of the specimens) decreased with the increasing temperature too. However, the ultimate strain was not affected as much as other parameters.

At 200 °C, in the rising stage of the stress–strain curves, the stress–strain curves of the three conditions were similar. The stress–strain curve of G200-0.4 was almost coincident with that at the room temperature. The stress–strain curve slope of G200 was slightly smaller than that of the other two curves. The residual compressive strength of mortar after being heated to 200 °C was decreased. The residual compressive strength of G200 was a little lower than that of G200-0.4. The peak strain of mortar after being heated decreased too and that of G200-0.2 was the smallest. The bearing capacity of C20 and G200 decreased sharply after the peak strength, the failure mode of these was a brittle failure. However, the bearing capacity of G200-0.2 decreased slowly after the peak strength.

At 400 °C, in the rising stage of the stress–strain curves, the stress–strain curves of the five conditions were similar and that of G400-0.4 was almost coincident with that at the room temperature. In the heated conditions, the stress–strain curve slope of G400-0.4 was the largest, and that of G400 was the lowest. The residual compressive strength of the mortar after being heated to 400 °C was similar and much lower than that at the room temperature. The peak strain of G400 was the largest and that of G400-0.4 was the smallest. The bearing capacity of G400-0.6 after the peak strength decreased much more slowly than the others.

At 600 °C, in the rising stage of the stress–strain curve, the stress–strain curves of G600 and G600-0.4 were similar, and the slopes of the two curves were much smaller than that of C20. The residual compressive strength of G600 and G600-0.4 decreased dramatically compared with that of C20, and the residual compressive of G600-0.4 was a little higher than that of G600. The peak strain of G600 and G600-0.4 was similar. The bearing capacity of G600-0.4 after the peak strength decreased faster than that of G600. The ultimate strain of G600 was the largest.

It can be seen in Figure 8e,f that, for the unstressed specimens, the residual compressive strength and the slope of the stress–strain curves decreased with the increasing temperature. Under the stressed conditions with the pre-stress ratio of 0.4, the same conclusion can be drawn on the residual compressive strength, however, the slopes of the stress–strain curves of the stressed mortar were almost never changed before 400 °C and decreased dramatically after 400 °C.

At 400 °C, the residual compressive strength of G400-0.4 was higher in the stressed conditions. The residual compressive strength of unstressed mortar was a little higher than that of the stressed mortar before 400 °C. However, at 600 °C, the stressed residual compressive strength was much higher than that under the unstressed. The residual compressive strength of mortar after being heated was shown in Table 2. Abrams [21] suggested that the difference within pre-stress ratios had insignificant effects on the compressive strength of specimens. Therefore, based on Guo et al. [44], the equations for the temperature dependence of the residual compressive strength of the mortar under unconfined and confined conditions were as follows and were shown in Figure 9.
(3)$$\frac{f^{T}}{f_{\max}}=1.015-0.93\frac{T}{1000}(R^{2}\;=\;0.967)$$
(4)$$\frac{f^{T\delta }}{f_{\max}}=\frac{1}{1+1.2(\frac{T}{1000}{)}^{0.37}}(R^{2}=0.88)$$
where $f_{\max}$ is the compressive strength of mortar at room temperature, $f^{T}$ is the residual compressive of cement mortar under the unstressed condition. $f^{T\delta }$ is the residual compressive of cement mortar under the stressed condition.

## 4. Discussion

The reason of the improvement in compressive strength of stressed concrete is still confusing. Khoury et al. [35] and Fu [29] argued that, due to the preload in the stressed condition, there was a ‘densification’ in the cement mortar, resulting in a reduction in porosity relative to unstressed condition. Therefore, the compressive strength under constrained conditions was higher than that under unconstrained conditions. However, a comprehensive analysis of the residual compressive strength test results and porosity test results of the cement mortar heated under different stress ratios showed that although the porosity of the test block under constrained conditions was slightly lower than that under unconstrained conditions, the conclusion that the increase in compressive strength was caused by the ‘denaturation’ of the cement slurry was not significant or even opposite.

At 200 °C, the compressive strength of unstressed specimens was higher than that of stressed specimens, but the porosity of unstressed mortar specimens was lower than that of stressed specimens. At 400 °C, both the strength and porosity of the mortar under the two conditions were similar. However, at 600 °C, the compressive strength of the unstressed specimens was lower than that of the stressed specimens, and the porosity of unstressed was higher than that of stressed specimens, which was opposite to that at 200 °C. Therefore, the ‘densification’ of the cement paste led to the increase in the compressive strength of stressed cement mortar was insufficient.

Many factors lead to the decrease of compressive strength as concrete exposure to high temperature [7]. As we have known, the thermal expansion of cement paste and aggregate were different. It can be seen that in Figure 10 [45], the higher the temperature, the greater the difference in thermal expansion. The experimental investigation carried out by Fu et al. [30] showed that mortar had a higher reduction rate of mechanical properties than hardened cement paste (HCP) at high temperature due to the thermal expansion mismatch between HCP and fine aggregate.

At 200 °C, the difference of the thermal expansion between cement paste and aggregate was small, and the decomposition of cement paste hydration was not serious, so the preload increased the damage of cement mortar at this temperature. With the increasing temperature, the difference of the thermal expansion between cement paste and aggregate increased, the decomposition of cement paste hydration, especially after 400 °C, was serious. However, the preload reduced the cracks, especially the cracks induced due to the thermal mismatch between the aggregate and HCP, which might be the reason that the compressive strength of stressed cement mortar increased after 400 °C. Clear cracks at G400 can be seen, but no crack was observed at G400-0.4 in the SEM image in Figure 11.

## 5. Conclusions

In this study, a series of tests were carried out on stressed mortar, the porosity of the mortar was tested by X-ray CT. The residual compressive strength of stressed mortar cooled to room temperature was measured. Based on the results of the tests, the following conclusions were drawn:

The porosity of cement mortar increased with the increasing temperature. The difference between the stressed and unstressed mortar was little, the preload loaded during heating has little effect on the porosity.

The preload loaded during heating has more effect on the residual compressive strength of mortar. The residual compressive strength of unstressed mortar decreased with the increasing temperature continually. However, the reduction of mortar residual compressive strength under constrained conditions was mainly after 400 °C.

Though the porosity of the stressed mortar was a little lower than that of the unstressed mortar, the ‘densification’ of the cement paste lead to the increase in the compressive strength of stressed cement mortar was insufficient. The preload reduced the crack, especially the crack induced due to the thermal mismatch between aggregate and HCP may be the main reason of compressive strength increasing.

## Figures and Tables

**Figure 1 materials-14-01944-f001:**
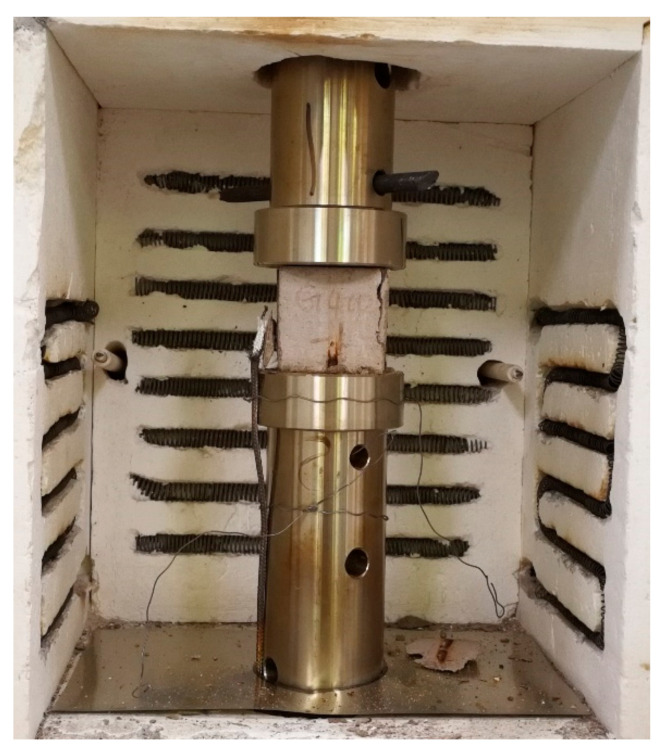
Test instrument.

**Figure 2 materials-14-01944-f002:**
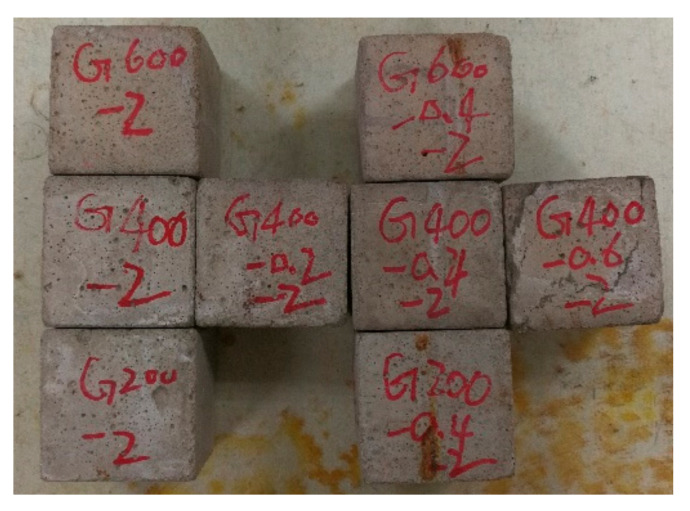
Mortar specimens after thermal treated.

**Figure 3 materials-14-01944-f003:**
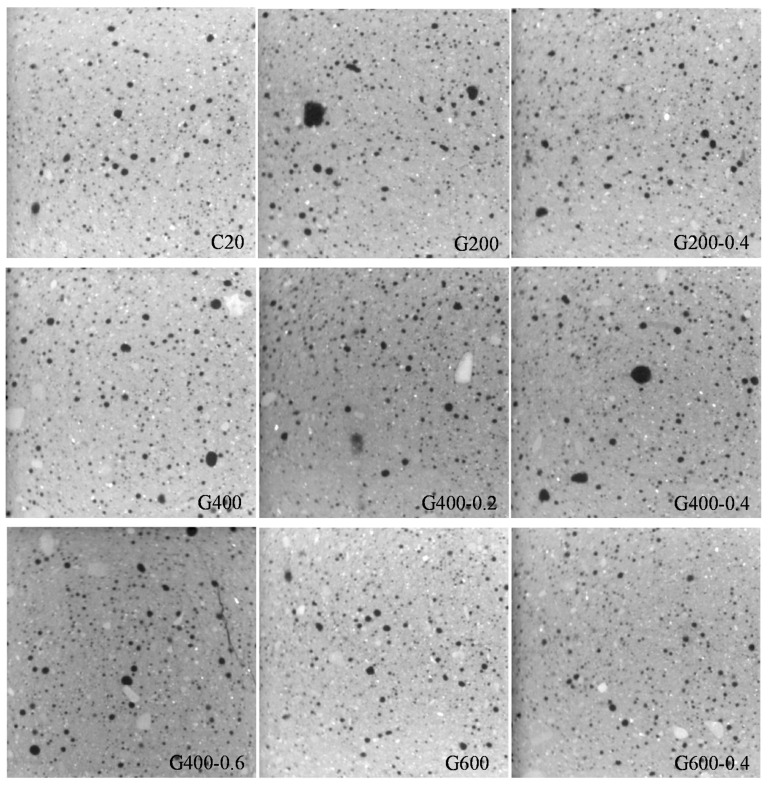
CT image of the middle cross section in X direction.

**Figure 4 materials-14-01944-f004:**
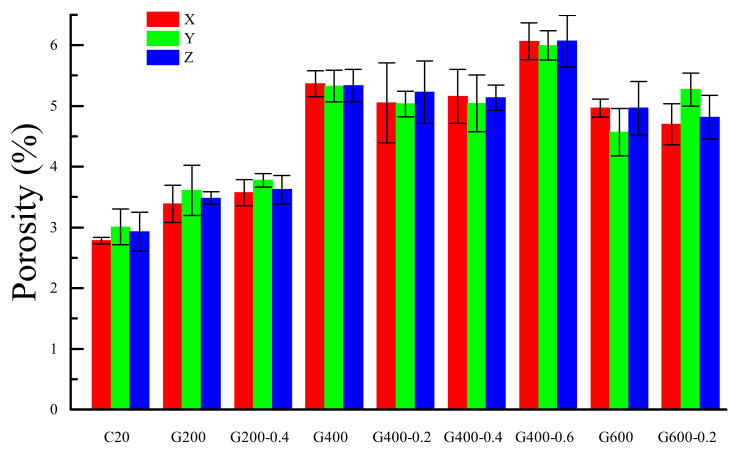
Average *D*_s_ in each direction.

**Figure 5 materials-14-01944-f005:**
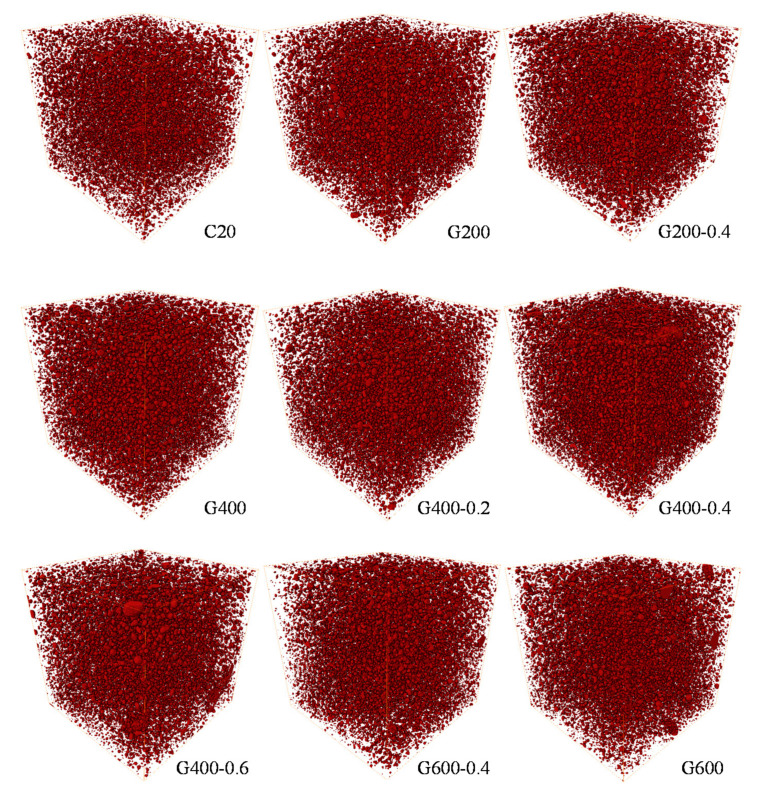
3D reconstruction images.

**Figure 6 materials-14-01944-f006:**
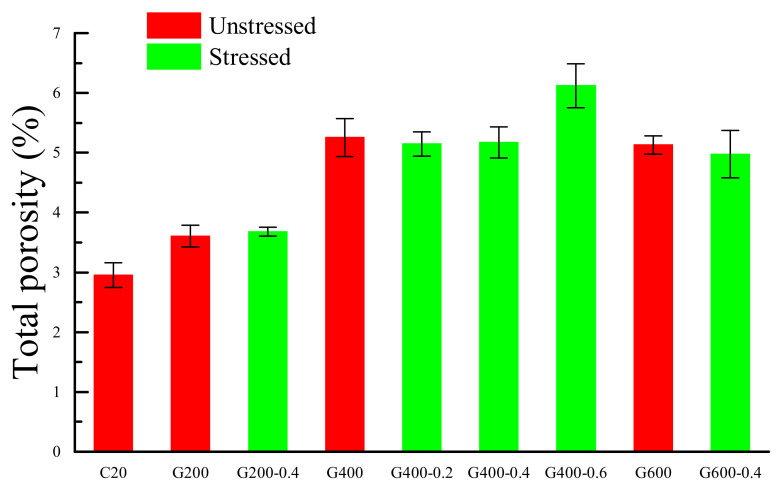
*D*_v_ of cement after thermal treated.

**Figure 7 materials-14-01944-f007:**
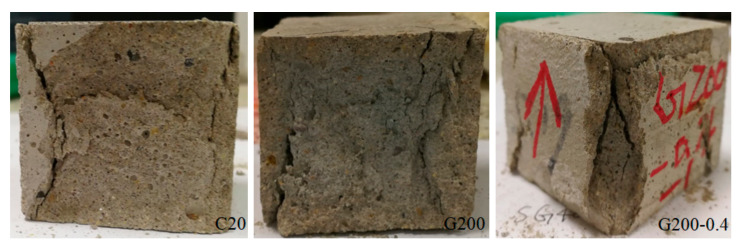
Mortar specimens after compressive strength test.

**Figure 8 materials-14-01944-f008:**
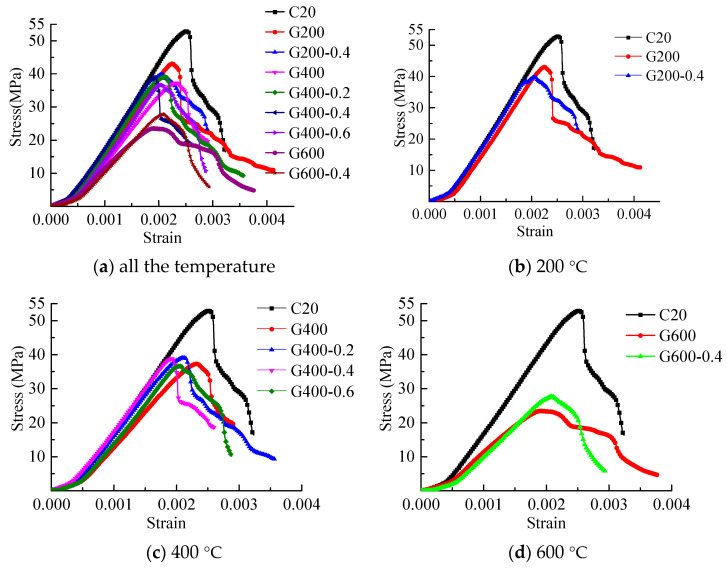
Compressive stress–strain curves of mortar after high temperature. (**a**) all the temperature; (**b**) 200 °C; (**c**) 400 °C; (**d**) 600 °C; (**e**) Unstressed; (**f**) Stressed.

**Figure 9 materials-14-01944-f009:**
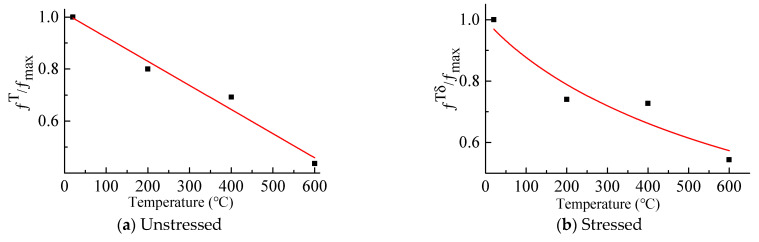
Fitting curves of the heated strength. (**a**) Unstressed; (**b**) Stressed.

**Figure 10 materials-14-01944-f010:**
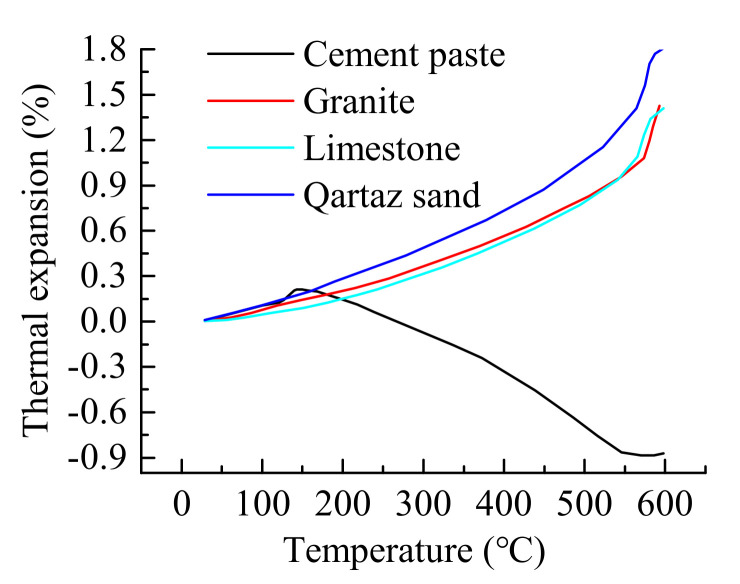
Thermal expansion of concrete compositions.

**Figure 11 materials-14-01944-f011:**
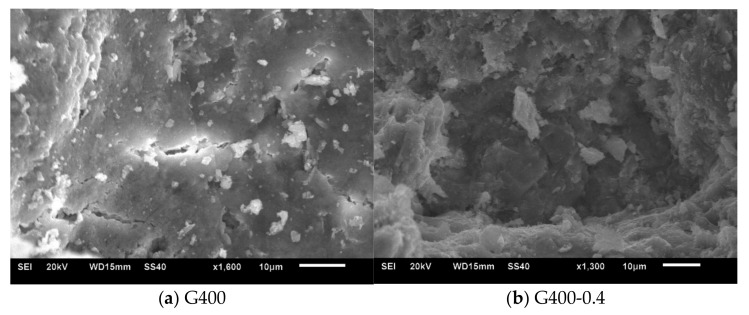
SEM images of mortar after thermal treated. (**a**) G400; (**b**) G400-0.4.

**Table 1 materials-14-01944-t001:** Test conditions.

Temperature/°C	Pre-Stress Ratio			
20	0	-	-	-
200	0	-	0.4	-
400	0	0.2	0.4	0.6
600	0	-	0.4	-

**Table 2 materials-14-01944-t002:** Residual compressive strength of cement mortar.

Temperature (°C)	Pre-Stress Ratio			
	0	0.2	0.4	0.6
20	53.8	-	-	-
200	43.07	-	39.8	-
400	39.56	38.2	39.4	38.56
600	23.45	-	29.25	-

## Data Availability

The data presented in this study are available on request from the corresponding author.

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
