# Peer review of "Investigation of the Residual Mechanical and Porosity Properties of Cement Mortar under Axial Stress during Heating"

_materials, 2021, doi:10.3390/ma14081944_

Round 1
Reviewer 1 Report
The current paper present an experimental study on the compressive strength of cement mortars exposed to high temperatures in order to determine the effect of the stress level of the samples when heating. The authors use x ray CT scan to determine the changes in porosity to explain differences on the mechanical behavior of the material. The topic could fit in the journal, and the methodology could be interesting. However, in my opinion, there are some problems to address before considering the manuscript for publication.
One of the problems regarding the microstructural changes in concrete at high temperatures is the different thermal expansion of each material, as the authors explain in the text. In this regard, coarse aggregates could have an important impact on the results. Why did you choose to use only fine aggregates? I assume that the study of microstructure could be easier.
How many specimens were prepared? It seems that only one of each condition was tested, and there are some rare results to rely in only one experimental measure. If more than one were tested, please included dispersion data when available.
The main decomposition in concrete starts between 400°C and 500°C, in which sometimes explosive spalling may appear due to excessive pore pressure. This deterioration is reflected in the loss of compressive strength in the 600°C specimen. Shouldn’t it be any changes in the porosity? According to figures 4 and 6 the porosity after 600°C is even lower than after 400°C.
How was the exposure time selected? May be it was too short and the damage at 400°C is not so severe (sometimes the strength was very close to 200°C).
Please, explain briefly how were the compressive tests performed, loading rate (force or displacement controlled), strain measurements. Can the elastic moduli be calculated? If so, please discuss the effect on the moduli instead of the slope of the stress-strain curve.
The trends obtained were similar to the previous studies included in the references?
When presenting the porosity results, please include the orientation of XYZ directions, with respect to the loading direction.
Line 200, what is areal voxel number?
In figure 3 only the G400-0.6 seems to be cracked, if several samples were tested, was this behavior repeated? Or could it be a fabrication defect?
How many data are included in figure 9? If there are only four points on each graph, I’m not sure about the statistical relevance of the given equations.
Fig. 11, the scale of (b) is twice the scale of (a), and it seems to be some cracks in the center of the image. Why are they different situations? Could SEM be related somehow to the CTs before?
Reviewer 2 Report
The presented results are interesting; however, some major issues should be addressed prior to the publication.
1) The biggest drawback of the work is the lack of any statistical analysis of the obtained test results. The number of analyzed samples (only 3) is insufficient to obtain a reliable test result for compressive strength measurements. In my opinion, the lack of an appropriate statistical test disqualifies this work in its present form.
2) No standard was referenced in the article. Please provide the standard number according to which the cement mortar samples were prepared and compressive strength tests were carried out. What does "clean water" mean (line 108)? Do the authors mean tap water or distilled water, for example?
3) Producers of cement and equipment used for samples testing (eg the compressive strength testing machine, HR X-CT, SEM microscope) were not given.
4) The graphs were made a bit carelessly: there are no units (Fig. 8), the axis description is incorrect (Fig. 9a), the scale is illegible (Fig. 4 and Fig. 10).
5) It is difficult to compare SEM photos as they are for different magnifications (x1600 and x850). In addition, cracks may have formed during the compressive strength tests.
Round 2
Reviewer 2 Report
Thank you, all doubts were clarified by the authors. Measurement errors have been marked on the graphs.